# InstanceEasyTL: An Improved Transfer-Learning Method for EEG-Based Cross-Subject Fatigue Detection

**DOI:** 10.3390/s20247251

**Published:** 2020-12-17

**Authors:** Hong Zeng, Jiaming Zhang, Wael Zakaria, Fabio Babiloni, Borghini Gianluca, Xiufeng Li, Wanzeng Kong

**Affiliations:** 1School of Computer Science and Technology, Hangzhou Dianzi University, Hangzhou 310018, China; jivon@hdu.edu.cn (H.Z.); 182050179@hdu.edu.cn (J.Z.); lixiufengcn@hdu.edu.cn (X.L.); kongwanzeng@hdu.edu.cn (W.K.); 2Key Laboratory of Brain Machine Collaborative Intelligence of Zhejiang Province, Hangzhou 310018, China; 3Department of Mathematics-Computer Science, Faculty of Science, Ain Shams University, Abbassia, Cairo 11435, Egypt; wael.zakaria@sci.asu.edu.eg; 4Industrial NeuroScience Lab, University of Rome “La Sapienza”, 00161 Rome, Italy; gianluca.borghini@uniroma1.it

**Keywords:** Electroencephalogram (EEG), InstanceEasyTL, fatigue driving, cross-subject, transfer learning

## Abstract

Electroencephalogram (EEG) is an effective indicator for the detection of driver fatigue. Due to the significant differences in EEG signals across subjects, and difficulty in collecting sufficient EEG samples for analysis during driving, detecting fatigue across subjects through using EEG signals remains a challenge. EasyTL is a kind of transfer-learning model, which has demonstrated better performance in the field of image recognition, but not yet been applied in cross-subject EEG-based applications. In this paper, we propose an improved EasyTL-based classifier, the InstanceEasyTL, to perform EEG-based analysis for cross-subject fatigue mental-state detection. Experimental results show that InstanceEasyTL not only requires less EEG data, but also obtains better performance in accuracy and robustness than EasyTL, as well as existing machine-learning models such as Support Vector Machine (SVM), Transfer Component Analysis (TCA), Geodesic Flow Kernel (GFK), and Domain-adversarial Neural Networks (DANN), etc.

## 1. Introduction

In recent years, there has been rapid increase in the number of traffic accidents, yielded to huge losses to people’s lives and their properties. A lot of evidence shows that driving under the condition of fatigue state (fatigue driving) is one of the main causes of traffic accidents. Statistical results also indicate that fatigue driving leads to 35–45% of road traffic accidents [1,2,3], and directly causes 1550 deaths, 71,000 injuries, and $12.5 billion in economic losses each year according to the reports of American National Highway Traffic Safety Administration (NHTSA) [4]. Therefore, it is of vital importance to design an efficient and accurate analysis model for detecting fatigue over time during driving.

Generally, there are three ways to detect fatigue mental states. The first is video-based detections. Computer vision techniques are used for detecting fatigue by analyzing the facial expressions such as blinking, eye closure duration, yawning, and so on [5]. In this way, the blink frequency is one of the key factors for detecting fatigue. However, changes in illumination or wearing sunglasses will reduce the detection effect, because such methods adopt the vertical and horizontal projection intensities of the image to detect whether blinking or not. The second is psychology-based surveys. The subjects evaluate their mental states by filling in some psychological questionnaires [6,7]. Such questionnaires rely heavily on the subjective factors of the subjects that maybe lack objective judgment for verification [8,9]. The third is the measurements based on physiological signals. Usually, EEG, electrooculogram (EOG), electrocardiogram (ECG) or their mixtures are employed for measuring or detecting brain activity, eye movements [10,11,12], etc. Thus, fatigue states could be correspondingly identified as well. EEG-based methods are considered to be one of the most efficient methods since EEG could directly reflect the activities of the brain’s nerve cells to reveal the changes of the drivers’ mental states during driving [13,14].

Recent studies have shown that EEG-based methods could achieve better reliability and accuracy compared with other methods [15]. So far, many classical machine-learning algorithms such as SVM [16,17], linear discriminant analysis (LDA) [18], K Nearest Neighbor (KNN) [19], etc., as well as some typical deep-learning models such as Long Short-Term Memory Network (LSTM) [20], Convolutional Neural Network (CNN) [21], are adopted to analyze and uncover those important patterns of EEG signals. Despite the great success of these methods, regarding the significant differences of EEG signals, many manually labeled EEG data is still required to perform the prediction and classification tasks of mental states, especially for cross-subject prediction, and a large amount of training is needed as well to identify those EEG patterns between different subjects. In other words, the existing analysis is to a certain extent not only time-consuming, but also subject-dependent, which will not be quite fit for cross-subject EEG analysis, and will decrease the classification performance for across subjects.

Transfer-learning models are efficient methodologies that aim at transferring the previously extracted features from a labeled domain to a similar but different domain with limited or no labels to perform some specific decision tasks on the different domain [22,23,24]. Transfer-learning methods have achieved good performance in areas of image recognition and classification. Easy Transfer Learning (EasyTL) [24] was proposed by Wang et al. in 2019 for image recognition, which is divided into two phases: the first one aligns both the source and target domains to reduce the differences between both domains, and the second one makes use of a kind of probability annotation matrix for classification [24]. Since EasyTL does not require model selection and hyper-parameter tuning, it could achieve better competitive performance.

Although EasyTL is easy, accurate, efficient, and extensible, due to the significant differences of EEG signals across subjects which will cause different distributions between EEG feature patterns for different subjects, it is still difficult for EasyTL to address the issue of EEG-based cross-subject fatigue mental state detection. Please note that cross-subject fatigue detection is to train a model by extracting EEG features of some subjects and using them in the analysis of mental states of other subjects. To overcome the above shortcoming of EasyTL in dealing with cross-subject EEG signals, in this paper, we propose an improved EasyTL-based method, InstanceEasyTL, to perform cross-subject EEG-based fatigue detection. To verify our proposed model, we also conduct the comparative analysis between InstanceEasyTL, SVM, and some other existing transfer-learning methods such as Transfer Component Analysis (TCA) [25], Geodesic Flow Kernel (GFK) [26], and Domain-adversarial Neural Networks (DANN) [27].

The paper is organized as follows: Section 2 introduces the setup of the fatigue driving experiment, including subjects, EEG recording and preprocessing, as well as EEG feature extraction. Section 3 presents the existing EasyTL method and states in detail our proposed InstanceEasyTL model. Section 4 shows the related experimental results. Section 5 compares and discusses these results with other existing models, and demonstrates the efficiency of the proposed method. Finally, Section 6 concludes our works.

## 2. Materials

### 2.1. Subjects

This experiment is performed under the approval of the local ethical committee of University of Rome “La Sapienza” (Rome, Italy). 15 subjects (ages range [24–30] years, mean ± std = 26.8 ± 3.2 years) are selected to participate in the driving experiment. All the subjects have qualified driving licenses. Before the experiment, all the subjects are informed and interpreted of the experimental intention, and sign the written consents. In addition, all of them are not allowed to drink alcohol the day before the experiment, and no caffeine is taken 5 h before the experiment, respectively. Finally, the experiment is conducted in accordance with the principles outlined in the Helsinki Declaration of 1975, which was revised in 2013.

### 2.2. Experimental Protocol

To eliminate the possible effects of circadian rhythms and meals, the experiment is performed between 2 and 5 pm. In addition, the Driving track is the Spa Francorchamps (Belgium) track, and the Vehicle type is an Alfa Romeo Giulietta QV (1750 TBi, 4 cylinders, 235 HP) on our driving simulation platform.

Figure 1 is the driving experimental setting. The experiment is started in a quiet environment without any noise, the subjects are asked to sit on a comfortable sofa to drive a car by controlling a steering wheel according to the stimulation track 1m in front of them.

There are 8 tasks during the experiment, as shown in Table 1. To obtain a reference, we define warm-up (WUP) as the baseline of the experiment that is at the beginning of the experiment. In WUP, the subject is asked to drive a car for 2 laps without any stimuli and any errors. Next stage performance (PERFO) is similar to WUP, but the total time of driving is 2% less than WUP. Then the task of “alert“ with video and “vigilance” with audio (TAV) is designed and appeared randomly during the process of the experiment to make the subjects feel tired more easily, which are follow after PERFO with a pseudo-random order, TAV3, TAV1, TAV5, TAV2, TAV4. Please note that they will receive visual or sound stimuli with different frequencies in TAVs stages. Different stimulus frequencies of “alert” or “vigilance” represent the different degree of stimulation, defined as TAV1, TAV2, TAV3, TAV4, and TAV5 with different stimulus intervals [28,29,30,31]. The TAVs duration will be set up depending on the total time spent in WUP. From TAV1 to TAV5, the stimuli intervals are 9800–10,200, 7700–8100, 5900–6300, 4100–4500, and 2300–2700 ms, respectively. The last stage is drowsiness (DROW) with a slow speed just like driving in a crowded city center, without any video or audio stimuli alongside the track for 2 laps. The subject should press “Button#1” with his left finger when ‘X’ appears, which is an “alert” task, and press “Button#2” with his right finger when two consecutive “beep”’s come, which is a “vigilance” task [32,33,34]. “Alert” stimuli is used to mimic the actual road conditions, such as traffic lights, pedestrians crossing the road, other vehicles, etc. (as shown in Figure 2) [35], while “vigilance” stimuli is used to stimulate the car radio, engine noise or a phone call (as shown in Figure 3). Usually, the whole experiment will last 2 h. WUP, PERFO, 5 TAVs, and DROW will need about 10 min, 2% less than that of WUP, 1 h, and 20 min, respectively. There is a break of about 5 min between each stage.

All subjects are required to take a half-hour training session to familiarize themselves with the simulator’s commands and interfaces before starting the experiment, the experimental flow is shown in Figure 4.

After finishing every stage, the subjects are required to fill in the NASA-TLX questionnaire to provide subjective workload perception during the task. Moreover, a run sheet is used to take note of the subjects’ drive performance (“off-road”) during the experiment. “off-road” means that the subject has driven out of the track at least with one wheel.

### 2.3. EEG Recording and Preprocessing

EEG is recorded by using a digital monitoring system (Brain Products GmbH, Germany), in which all 61 EEG channels are referenced to the earlobe, grounded to FCz, their impedances are kept below 10 K Ω, and the sampling frequency F_s is 200 Hz. We use eeglab toolbox in Matlab to pre-process and process EEG data. A band-pass filter is employed to keep EEG signals with a frequency range between 1 and 30 Hz for fatigue driving analysis [36], independent component analysis (ICA) [28,37] is used for removing EOG artifacts.

### 2.4. EEG Feature Extraction

Before feeding the EEG data of each subject into the classifier for training, we extract the power spectrum density (PSD) features from EEG data of each subject [38,39], which is usually used for feature extraction in EEG analysis. The detailed PSD feature exaction procedure is listed as the following steps:For the recorded EEG of each channel, a 0.5 s hamming window without overlapping between two successive windows is used for dividing EEG into multiple samples. We extract 1400 hamming windows of sample points and each window have 0.5 s × F_s = 0.5 × 200 = 100 sample points (as shown in Figure 5a). Thus, the number of hamming windows (HW) is 1400 and the sample points in each hamming window are 100 × 61 channels = 6100.For each channel in each window, we apply the one-sided PSD-estimate of EEG signals with the frequency of 200 Hz that represent the strength in terms of the logarithm of power content of the signal at integral frequencies between 0 and 100 Hz. This produces 101 feature-length vectors. Therefore, 100 sample points in each channel become 101 features as well (as shown in Figure 5b).From the acquired 101 features in step 2, we can then extract 27 features at the frequency bands of θ band (4–7 Hz), α band (8–13 Hz), and β band (14–30 Hz), respectively [40] (as shown in Figure 5c).Then, the extracted features will be appended together to form D = (61 × 27) = 1647 dimension of feature vectors (as shown in Figure 5d).Consequently, for those HW =1400 windows/samples, we now have a feature space FS with HW × D =1400 × 1647 of order that will be fed into our proposed model for training (as shown in Figure 5e).

## 3. Methods

### 3.1. The Existing EasyTL Method

EasyTL [24] is a kind of transfer-learning method that has been applied in the field of image applications and achieved better performance. It consists of two parts: intra-domain alignment and intra-domain programming, as shown in Figure 6. For intra-domain alignment, it aligns $n_{s}$ samples to form the sample set $\Omega_{s}$ in the source domain *s* as ${\left\{x_{i}^{s}\right\}}_{i=1}^{n_{s}}$, and $n_{t}$ samples to form the sample set $\Omega_{t}$ in the target domain *t* as ${\left\{x_{j}^{t}\right\}}_{j=1}^{n_{t}}$ through intra-domain alignment [41], aiming at making the difference between *s* and *t* as small as possible. Please note that the source domain is ${\left\{(x_{i}^{s},y_{i}^{s})\right\}}_{i=1}^{n_{s}}$ and the target domain is ${\left\{x_{j}^{t}\right\}}_{j=1}^{n_{t}}$.

Intra-domain programming builds the classifier model by proposing a new Probability Annotation Matrix *W*, the rows of *W* denote the class label c∈{1,2,…,C}, and the column $x_{j}^{t}$ represents the target samples. The element $W_{cj}$ indicates the annotation probability of $x_{j}^{t}$ belonging to class c. Based on the matrix *W*, EasyTL can predict the target samples. Please note that the class labels of the target sample $x_{j}^{t}$ that we choose are the corresponding ones with the maximum of $\left\{W_{cj}\right\},j\in \left\{1,2,\ldots ,n_{t}\right\}$. For instance, as shown in Figure 7, the class label of $x_{1}^{t}$ will belong to class $C_{3}$ since it has the maximum probability of 0.4 among all elements $\left\{W_{11},W_{21},W_{31},W_{41}\right\}$ with the probabilities of {0.2,0.3,0.4,0.1}, respectively.

### 3.2. InstanceEasyTL

The main reason EasyTL has a better performance in the field of image recognition is that there is basically not much difference on pixel features between the images in the target domain and those in the source domain, which makes it be relatively easy to adopt the intra-domain alignment to align the target domain with the source domain. However, for cross-subject EEG analysis, due to the significant differences of EEG among different subjects, which will lead to the large difference in the distribution of EEG features. The existing intra-domain alignment method in EasyTL is difficult to align the features between the source domain and target domain, hence it is difficult to obtain an ideal cross-subject analysis effect.

Therefore, in this paper, an improved EasyTL-based method, InstanceEasyTL, is proposed for overcoming such shortcoming of EasyTL for cross-subject EEG analysis. The main idea of the proposed InstanceEasyTL lies in the aspect that to match the different distribution of EEG signals from different subjects, we adopt a strategy of alignment with certain weights to align EEG samples collected from both source and target domains. To achieve this goal, InstanceEasyTL will “borrow” some EEG samples from the target domain $\Omega_{t}$, together with the original source domain $\Omega_{s}$ (also called $T_{sd}$) to form a new sample set of source domain for training, which will get more EEG data and reduce the cost of EEG collection. As shown in Figure 8, the initial target domain $\Omega_{t}$ is divided into two parts: *S* and $T_{td}$, in which $T_{td}$ is employed as part of the new source domain $\Omega_{s}^{{}^{\prime }}$, thus $\Omega_{s}^{{}^{\prime }}$ consists of the initial source domain $T_{sd}$ and $T_{td}$ in $\Omega_{t}$, and accordingly, the new target domain $\Omega_{t}^{{}^{\prime }}$ only includes the part of *S*.

Mathematically, InstanceEasyTL can be depicted as follows.

First, we can determine $T_{sd}$ and $T_{td}$ according to the coefficient λ as Equations (Equation 1) and (Equation 2), respectively:(1)$$T_{sd}=\left\{\left(\Omega_{s},y_{s}\right)\right\}={\left\{\left(x_{i}^{sd},y_{i}^{sd}\right)\right\}}_{i=1}^{n_{s}}$$
and
(2)$$T_{td}=\left\{\lambda \left(\Omega_{t},y_{t}\right)\right\}={\left\{\left(x_{i}^{td},y_{i}^{td}\right)\right\}}_{i=1}^{m}$$
where $y_{s}$ and $y_{t}$ are the sets of class labels corresponding to $\Omega_{s}$ and $\Omega_{t}$. ($x_{i}^{sd}$, $y_{i}^{sd}$) and ($x_{i}^{td}$, $y_{i}^{td}$) are the i-th sample and its corresponding class label in the source domain $\Omega_{s}$ and target domain $\Omega_{t}$, respectively. $n_{s}$ and *m* are the number of samples in $T_{sd}$ and $T_{td}$.

Accordingly, we can then form the new source domain $\Omega_{s}^{{}^{\prime }}$ by Equation (Equation 3), and the new target domain $\Omega_{t}^{{}^{\prime }}$ by Equation (Equation 4), respectively:(3)$$T=\{T_{sd}\cup T_{td}\}$$
and
(4)$$S=\left(1-\lambda \right)\Omega_{t}={\left\{x_{j}^{t}\right\}}_{j=1}^{l}$$
here, T and S denote the sample sets of $\Omega_{s}^{{}^{\prime }}$ and $\Omega_{t}^{{}^{\prime }}$, respectively, *l* is the number of samples in $\Omega_{t}^{{}^{\prime }}$.

Algorithm 1 illustrates the proposed InstanceEasyTL method in detail.

First, initial weights are assigned to both training source domain $T_{sd}$ and training target domain $T_{td}$ by using Equation (Equation 5). We should note that compared with the EasyTL method, $T_{sd}$ = $\Omega_{s}$, $T_{td}\subset$
$\Omega_{t}$, $m<n_{t}$, and $m+l=n_{t}$, respectively (can also refer to Figure 8).

Secondly, the assigned weights for both $T_{sd}$ and $T_{td}$ are divided by the summation of all weights and stored as $p^{t}$ (as shown in Equation (Equation 6)). Based on the intra-domain programming method (also called EasyTL(c) in Algorithm 1), we take the training sample set *T* in $\Omega_{s}^{{}^{\prime }}$ (please see Equation (Equation 3)), $p^{t}$, and the new testing set *S* in $\Omega_{t}^{{}^{\prime }}$ (please see Equation (Equation 4)) as the input of InstanceEasyTL algorithm, then we can calculate the output $h_{t}$ of EasyTL(c). Here, *S* is not for the update of the weights, but for the testing after the end of iterations.

Thirdly, we will calculate the error $\epsilon_{t}$ between $h_{t}$ and real class labels y(x). The weights of $T_{td}$ and $T_{sd}$ are updated by $\beta_{t}$-based function (please see steps 6 and 7 in Algorithm 1).

Finally, if the number of iteration reaches *N*, the expected output in *S* will be calculated by Equation (Equation 10).
**Algorithm 1. **InstanceEasyTL **Require:** the labeled sample set *T* in $\Omega_{s}^{{}^{\prime }}$ (Equation (Equation 3)), the unlabeled test set *S* in $\Omega_{t}^{{}^{\prime }}$ (Equation (Equation 4)), the intra-domain programming method EasyTL(c) and the maximum number of iterations *N*.1:**Initialize:** the initial weights, calculated by Equation (Equation 5) and the initial number of iterations *t* = 1. Where $w_{sd}^{i}$ and $w_{td}^{i}$ are the weights of the i-th samples in $\Omega_{s}^{{}^{\prime }}$ and $\Omega_{t}^{{}^{\prime }}$. $W_{sd}^{1}$, $W_{td}^{1}$ and $W^{1}$ are the sets of weights in $\Omega_{s}^{{}^{\prime }}$, $\Omega_{t}^{{}^{\prime }}$ and both of source and target domains after one iteration, respectively.
(5)$$\begin{array}{c} W_{sd}^{1}=\overset{n_{s}}{\underset{i=1}{⋃}}w_{sd}^{i}\\ W_{td}^{1}=\overset{n_{s}+m}{\underset{i=n_{s}+1}{⋃}}w_{td}^{i}\\ W^{1}=W_{sd}^{1}\cup W_{td}^{1} \end{array}$$2:Set
(6)$$\begin{array}{c} {\mathrm{sum}}^{t}=\sum_{w\in W^{t}}w\\ p^{t}=\left\{\frac{w}{{\mathrm{sum}}^{t}};w\in W^{t}\right\} \end{array}$$3:After t∈{1,2,…,N} iterations, *w*, $W^{t}$ and ${\mathrm{sum}}^{t}$ represent the weight of one sample, the set of weights of all samples, and the sum of *w* in $\Omega_{s}^{{}^{\prime }}$ , respectively, and $p^{t}$ means the set of the weight *w* of each sample in $\Omega_{s}^{{}^{\prime }}$ in proportion of ${\mathrm{sum}}^{t}$.4:After *t* iterations, take *T*, $p^{t}$ and *S* as the input of EasyTL(c) to calculate the expected class label $h_{t}\left\{x\right\}$ of sample *x* in *T* and *S*.5:After *t* iterations, calculate the error $\epsilon_{t}$ of sample *x* between $h_{t}\left(x\right)$ and y(x) in $T_{td}$, here, y(x) returns the real label of sample *x* in $T_{td}$.
(7)ϵt=1Wt∗∑x∈Ttdwx∈Wtdtwx|ht(x)−y(x)|6:Define functions $\beta_{t}$ and β to update $W_{td}$ and $W_{sd}$, respectively. As shown in Equation (Equation 8) [42].
(8)$$\begin{array}{c} \beta_{t}=\epsilon_{t}/\left(1-\epsilon_{t}\right)\\ \beta =1/\left(1+{\left(2\ln n_{s}/N\right)}^{1/2}\right) \end{array}$$7:The new weight set after *t* iterations can be obtained by Equation (Equation 9).
(9)Wt+1=⋃x∈Tsdwx∈Wsdtwxβht(x)−y(x)∪⋃x∈Ttdwx∈Wtdtwxβt−ht(x)−y(x)8:*t* = *t* + 1, Go back to step 2 util *t* = *N* **Ensure:** the final expected class labels $h_{f}\left(x\right)$ in *S* according to Equation (Equation 10). herein, $h_{t}\left[x\right]$ is the expected class label of sample *x* in *S* for each step of the loop, which can be got from step 4:
(10)$$h_{f}\left(x\right)=\left\{\begin{array}{c} 1,\prod_{t=\lceil N/2\rceil }^{N}\beta_{t}^{-h_{t}\left[x\right]}\geq \prod_{t=\lceil N/2\rceil }^{N}\beta_{t}^{-\frac{1}{2}}\\ 0,\;\mathrm{otherwise}\; \end{array}\right.$$

## 4. Results

The experiment is performed on a GPU with a memory of 64 GB, Titan Xp graphic memory with 12 GB. Additionally, Intel i5-4570 CPU with a frequency of 3.2 G Hz, 1 TB of storage capacity, and 8 GB of memory are also employed to run these algorithms. InstanceEasyTL and other comparable models except for DANN are tested on MATLAB R2016b software and DANN on pycharm_2018.3.3 software. Codes are available at https://github.com/13etterrnan/InstanceEasyTL.

We compare InstanceEasyTL with the traditional machine-learning and transfer-learning methods, including SVM with linear kernel and intra-domain alignment [41], Transfer Component Analysis (TCA), Geodesic Flow Kernel (GFK), Domain-adversarial Neural Networks (DANN), and the existing EasyTL.

### 4.1. Selection of Experimental Conditions

To distinguish experimental conditions used to train models, we make use of the subjective (i.e., NASA_TLX) and subjects’ drive performance (i.e., “off-road”). With the change of the workload in different experimental conditions, it shows TAV3 and DROW have the highest and lowest workload, respectively. In addition, due to TAV3 is the first stage with audio and video stimuli in each experiment, subjects will be most sober. Correspondingly, when entering into the stage of DROW, the subjects are very likely to feel tired. Therefore, TAV3 and DROW are used as the typical mental states for analysis.

There are a total of 1400 samples for each subject, including 700 samples for TAV3 and 700 samples for DROW, respectively.

### 4.2. Parameter Configurations of Models

For SVM, The kernel type = linear. For TCA, the used parameters are λ=1, dim=30, γ=1, and kernel type = primal. For GFK, the used parameter is dim=20. For DANN, the used parameters are epoch=100, batch_size=64, learningrate=0.0005, and optimizer=adam. In InstanceEayTL, λ in Equation (Equation 2) is set to be 0.3.

In this paper, we set the number of loops N=30 for InstanceEasyTL, since we find that the performances reach convergence after 30 loops. We perform 15 times of experiments to test the performances of InstanceEasyTL, in each experiment, 14 of 15 subjects are used as training samples, and the remaining as the testing samples. Thus, after 15 experiments, each subject is employed as the target domain to be tested at least one time, and the $\Omega_{t}$ in each experiment belongs to a different subject to ensure we can acquire more objective performances of InstanceEasyTL.

### 4.3. Classification Accuracy of InstanceEasyTL

Figure 9 shows the classification performance of different models. We can find that in these 15 experiments, InstanceEasyTL has almost the highest classification accuracy in cross-subject EEG analysis among these classifiers, which is more obvious in the three experiments of Sub_1-others, Sub_3-others, and Sub_15-others. Regarding the average performance, the average accuracy of InstanceEasyTL is 88.33%, which is significantly higher than that of SVM (61.65%), TCA (58.01%), GFK (56.32%), DANN (70.75%), and EasyTL (70.91%). Please note that it is 15% higher than that of the second-ranked classifier.

### 4.4. Statistical Analysis Results

To further analyze the performance of InstanceEasyTL on fatigue driving detection, we calculate the indicators of Recall, Precision, and F1score for each classifier, the results are shown in Table 2. These three indicators can be calculated as follows:(11)$$\begin{array}{c} Precision=TP/(TP+FP)\\ Recall=TP/(TP+FN)\\ F1score=2\times \frac{Precision\times Recall}{\left(Precision+Recall\right)} \end{array}$$
where TP is the number of samples correctly predicted as DROW, FP is the number of samples incorrectly predicted as DROW, FN is the number of samples incorrectly predicted as TAV3.

Therefore, Recall means the ratio of the samples correctly predicted as DROW to total samples in DROW state, and Precision means the samples correctly predicted as DROW to the samples predicted as DROW, respectively, F1score is the harmonic mean of the precision and recall.

We can find in 15 experiments InstanceEasyTL outperforms all other methods. The average Recall, Precision, and F1score achieved by InstanceEasyTL on our dataset are 89.08, 88.02, and 88.46, respectively, which significantly outperforms DANN by 18%, 16%, and 17%, respectively.

### 4.5. The Impact of Different $T_{td}:\Omega_{t}$ on InstanceEasyTL

In this section, we count the impact of different ratio of $T_{td}:\Omega_{t}$ on InstanceEasyTL. Here, we set the ratio of $T_{td}:\Omega_{t}$ to be 0.1, 0.2, 0.3, 0.4, and 0.5, respectively. We can find that different ratio of $T_{td}:\Omega_{t}$ results in different classification accuracy on InstanceEasyTL, as shown in Figure 10. From Figure 10, the cross-subject recognition accuracy of InstanceEasyTL is above 80% and shows an increasing trend as the ratio of $T_{td}:\Omega_{t}$ increases, except for very few exceptions. Similar results can also be observed from Figure 10p by calculating the average accuracy of InstanceEasyTL. We can then conclude that the higher the ratio of $T_{td}:\Omega_{t}$, the higher the accuracy of InstanceEasyTL.

## 5. Discussion

Compared to other traditional methods, InstanceEasyTL could acquire better classification accuracy performance for cross-subject EEG analysis. There are mainly two reasons. First, from the perspective of data analysis, InstanceEasyTL “borrows” some samples and labels from the target domain $\Omega_{t}$, which will be regarded as a part of the new generated source domain $\Omega_{s}^{{}^{\prime }}$, thus, InstanceEasyTL can acquire more information of $\Omega_{t}$ to make classification accuracy much higher. However, for other traditional methods, although the samples in the target domain can also be used for classification, they usually have no corresponding labels for these samples. Secondly, from the perspective of model training, InstanceEasyTL adjusts the weights of samples during multiple iterations, and can adaptively choose the samples that are similar to those in the target domain $\Omega_{t}$, which is more helpful for the subsequent training process. Because InstanceEasyTL adopts the samples and labels from $\Omega_{t}$ for training, which comes from different subjects, much better performance across subjects will then be acquired. Although other methods, such as TCK and EasyTL, have proven to be effective in image recognition applications, because they make the feature distributions between the source and target domains much similar, due to the significant differences of EEG signals over time and across subjects, it is very difficult to train those models to acquire similar features between the source and target domains for EEG analysis across subjects.

Then, we employ Recall, Precision, and F1score for statistical comparison analysis. It can be known from Equation (Equation 11) that the greater the Recall/Precision/F1score are, the easier it can distinguish DROW from TAV3. From Table 2, we can find InstanceEasyTL has the highest values of Recall, Precision, and F1score statistical analysis results, which illustrates better classification performance that InstanceEasyTL has in the cross-subject analysis.

Moreover, the impact of different ratios of $T_{td}:\Omega_{t}$ on InstanceEasyTL is counted as well, as shown in Figure 10. Here, S and $T_{td}$ are used to form the new target domain $\Omega_{t}^{\prime }$ and new source domain $\Omega_{s}^{\prime }$, respectively. Accordingly, samples in $\Omega_{s}^{\prime }$ and $\Omega_{t}^{\prime }$ are used by InstanceEasyTL for training and testing. In general, the different ratios of samples in these two domains will also affect the performance of InstanceEasyTL. Obviously, as the ratio of $T_{td}:\Omega_{t}$ increases, on the one hand, the samples used for training in $\Omega_{s}^{\prime }$ get larger accordingly, the better the performance of InstanceEasyTL as well. On the other hand, in this way, InstanceEasyTL can extract more characteristics from the original target domain $\Omega_{t}$, which is more suitable for EEG analysis across subjects. Additionally, we can find more information from $\Omega_{t}$ when $T_{td}:\Omega_{t}$ equals to 0.7 since the number of samples in $\Omega_{s}^{\prime }$ is getting larger, which is more helpful to InstanceEasyTL for extracting the characteristics in $\Omega_{t}$, and helps improve the performance of InstanceEasyTL as well. However, few abnormal results exist. For instance, as shown in Figure 10a when the ratio of $T_{td}:\Omega_{t}$ increases from 0.3 to 0.4, the accuracy of InstanceEasyTL decreases. The main possible reason is that we do not completely remove the artifact components in EEG signals during preprocessing stage, which will result in the inclusion of some artifact components in the new source domain $\Omega_{s}^{\prime }$ or the new target domain $\Omega_{t}^{\prime }$, thereby reducing the classification accuracy of InstanceEasyTL.

## 6. Conclusions

In this paper, we propose an EasyTL-based model, named InstanceEasyTL, that focuses on performing fatigue detection across subjects based on EEG. InstanceEasyTL can extract important characteristic information from some subjects and then transfer it to the other new subjects by assigning weights to the samples and “borrowing” a part of data from the new subject as training samples. To verify the performance of InstanceEasyTL, we compare it with the traditional methods, such as SVM, TCA, GFK, DANN, and EasyTL. The results show that InstanceEasyTL can obtain better cross-subject classification performance. Moreover, the statistical analysis shows that InstanceEasyTL can better distinguish the mental state of DROW from TAV3, hence we can then make a further conclusion that InstanceEasyTL has a more comprehensive classification performance. In addition, by adjusting the ratio of $T_{td}:\Omega_{t}$, we can find the performance of InstanceEasyTL will be improved as the ratio increases. In our future work, we will continue to focus on the research on partial transfer learning where the target domain label space is a subspace of the source domain label space.

## Figures and Tables

**Figure 1 sensors-20-07251-f001:**
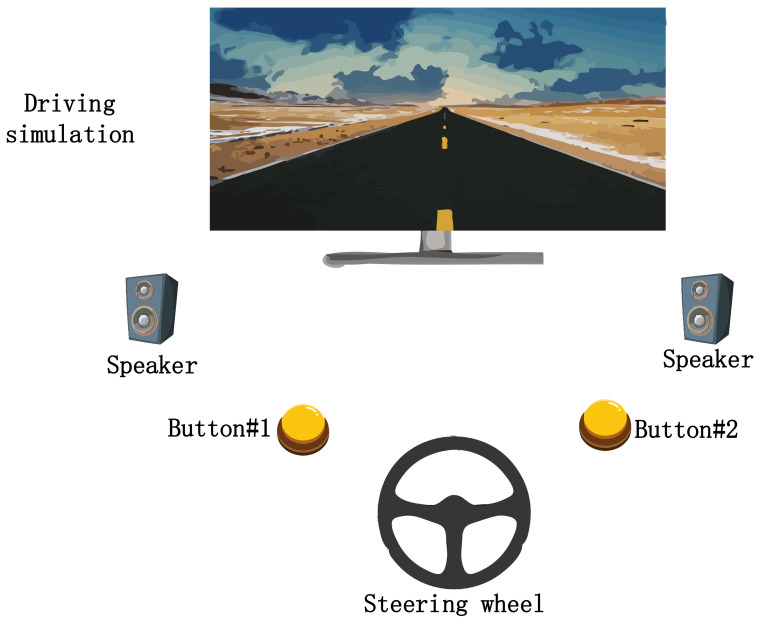
Experimental settings.

**Figure 2 sensors-20-07251-f002:**
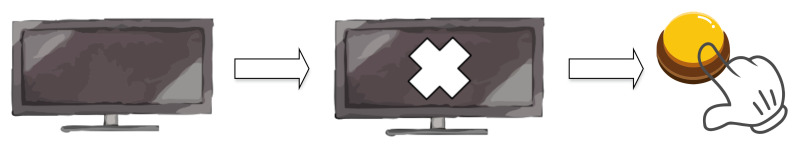
Alert stimuli response process in TAV.

**Figure 3 sensors-20-07251-f003:**
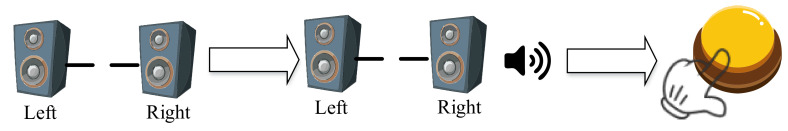
Vigilance stimuli response process in TAV.

**Figure 4 sensors-20-07251-f004:**
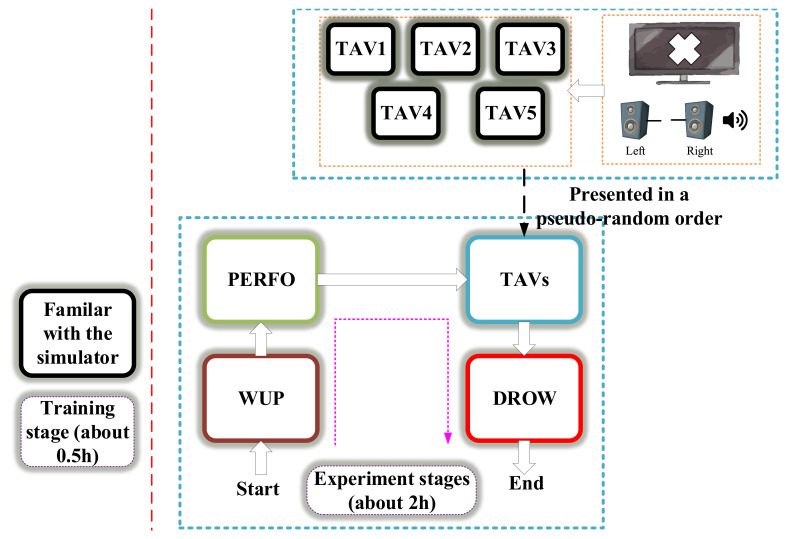
The experimental flow.

**Figure 5 sensors-20-07251-f005:**
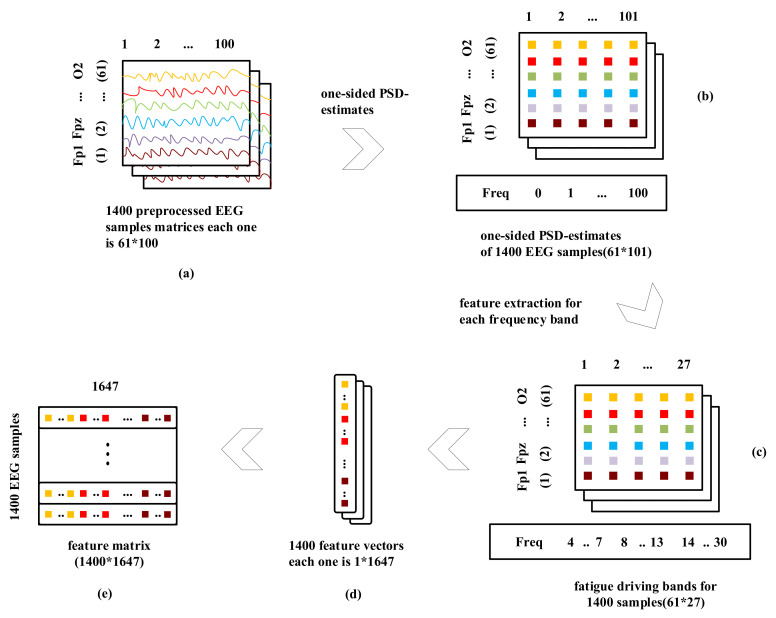
EEG feature vectors generation process.

**Figure 6 sensors-20-07251-f006:**
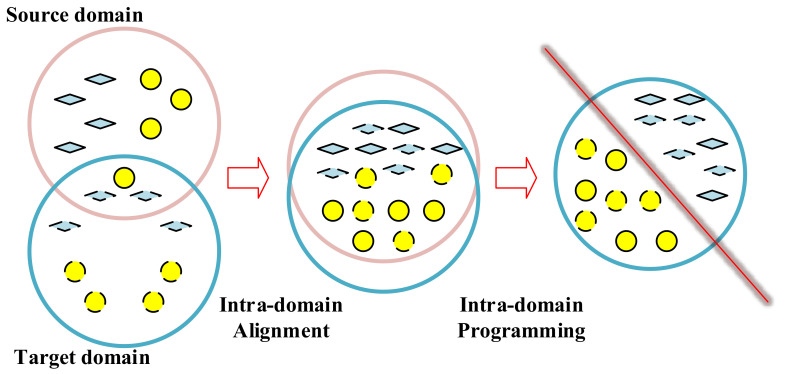
The procedure of EasyTL. The colored diamonds and circles denote samples from the source and target domain, respectively. The red line denotes the classifier.

**Figure 7 sensors-20-07251-f007:**
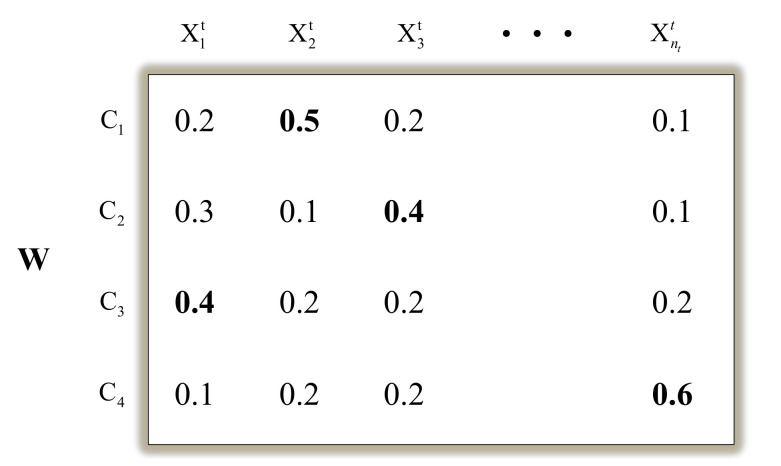
An example of the probability annotation matrix. The rows denote the class labels, and the columns denote the target samples. The entry value $W_{ij}$ indicates the annotation probability of $x_{j}^{t}$ belonging to class $C_{i};i=1,2,3,4$ and $j=1,2,\ldots ,n_{t}$. The class labels of $x_{j}^{t}$ belonging to class $C_{i}$ are marked in bold.

**Figure 8 sensors-20-07251-f008:**
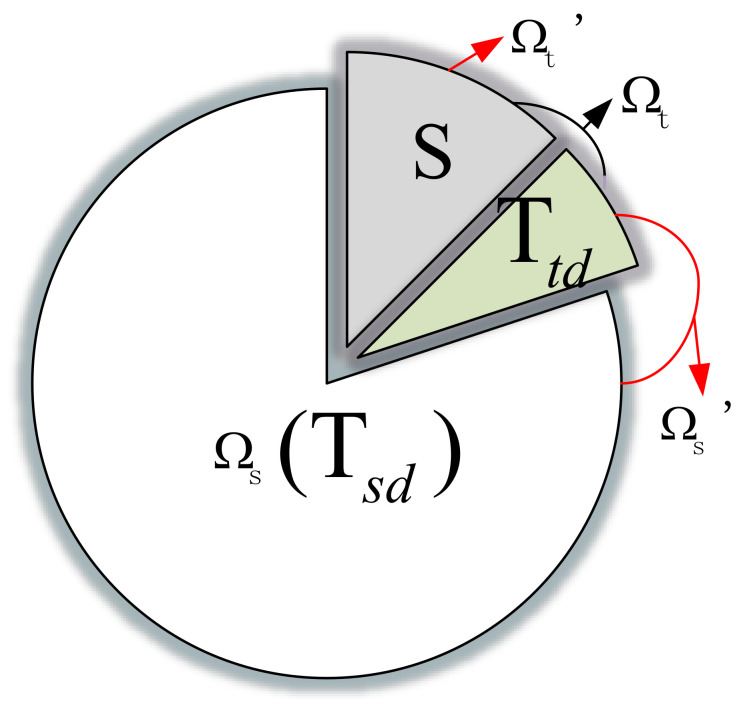
New training sample set in InstanceEasyTL.

**Figure 9 sensors-20-07251-f009:**
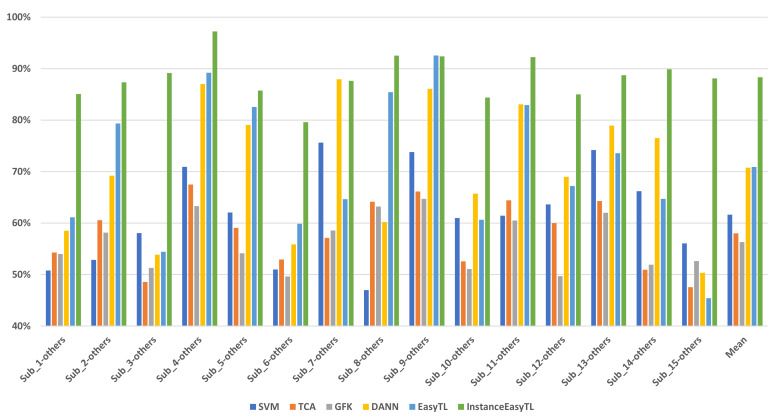
Classification accuracy comparison. X axis is represented as Sub_i-others (i = 1, 2, …, 15), where Sub_i is the samples of target domain, others are the samples of source domain. For example, the samples of “Sub_1” are used as the target domain, and those of the other 14 subjects are represented as “others” as the source domain. Y axis shows classification accuracy of InstanceEasyTL and other methods used for comparison.

**Figure 10 sensors-20-07251-f010:**
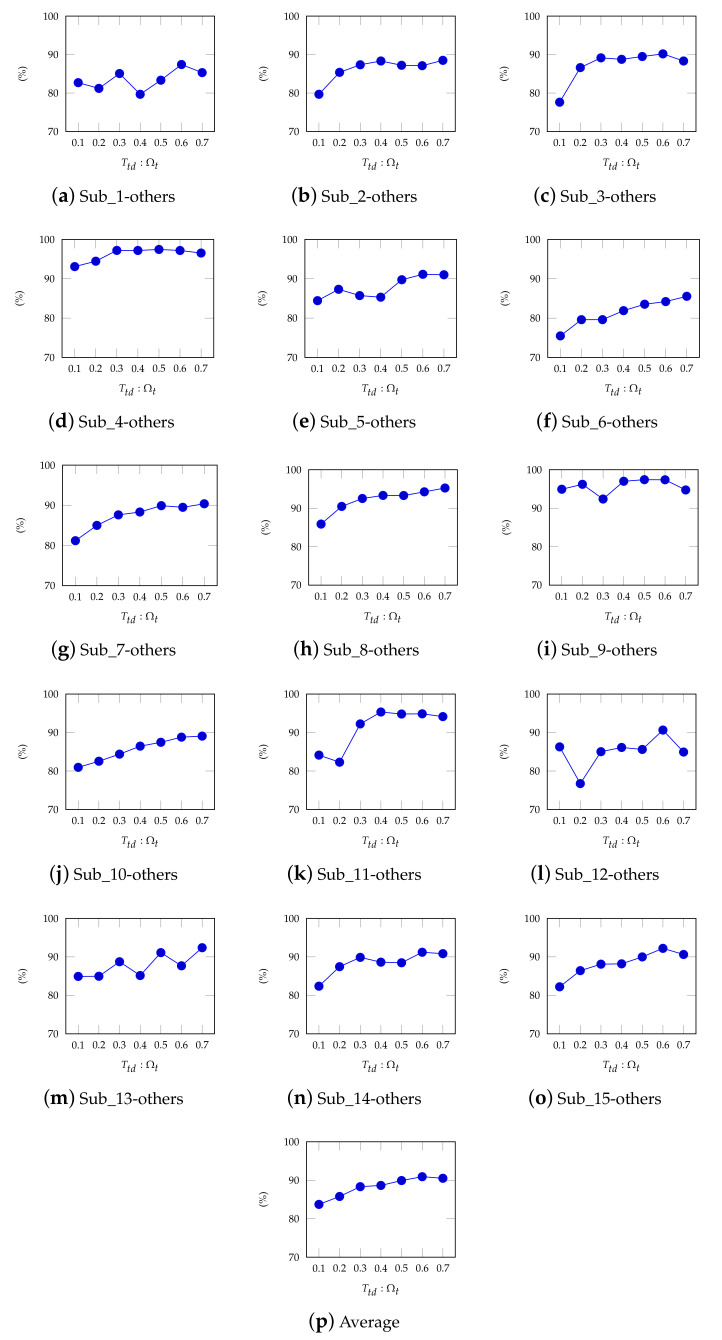
The accuracy of InstanceEasyTL with different $T_{td}:\Omega_{t}$, where Sub_i-others (i = 1, 2, …, 15) has the same meaning as in Figure 9. We also calculate the average accuracy of InstanceEasyTL.

**Table 1 sensors-20-07251-t001:** Description of different tasks.

WUP	WUP is as a baseline, which requires the driver to drive the car throughout the whole track without any speed requirements and any extra stimuli, but always kept the car in the path.
PERFO	In PERFO, the subject is asked to improve his previous performance of 2%. (total time = baseline − 2%)
TAVs	Five TAVs (TAV1 to TAV5) are presented in a pseudo-random order, TAV3,TAV5,TAV1,TAV2,TAV4,
	where TAV1 is in the situation with the longest stimulus interval, TAV5 is in the situation with the
	shortest stimulus interval.
DROW	DROW is the stage that the subject is required to drive slowly without any requirements on the speed.

**Table 2 sensors-20-07251-t002:** Recall, Precision and F1score(%) performance.

Target Domain (Source Domain)	SVM			TCA			DANN			EasyTL			InstanceEasyTL		
	Recall	Precision	F1score	Recall	Precision	F1score	Recall	Precision	F1score	Recall	Precision	F1score	Recall	Precision	F1score
S1 (S2–S15)	50.85	50.79	48.85	61.14	53.77	57.22	57.97	61.86	59.85	54.43	62.87	58.35	83.34	86.37	84.77
S2 (S1,S3–S15)	53.25	52.86	49.85	66.00	59.54	62.60	67.44	74.29	70.70	73.71	83.09	78.12	90.71	84.80	87.64
S3 (S1,S2,S4–S15)	59.22	58.07	55.29	48.43	48.57	48.50	53.65	56.71	55.14	53.57	54.51	54.03	88.86	89.45	89.14
S4 (S1–S3,S5–S15)	71.26	70.93	70.70	69.14	66.94	68.03	84.81	90.14	87.40	85.86	92.04	88.84	96.61	97.82	97.20
S5 (S1–S4,S6–S15)	62.23	62.07	61.83	60.43	58.83	59.62	79.28	78.71	79.00	80.71	83.83	82.24	86.97	84.77	85.84
S6 (S1–S5,S7–S15)	51.11	51.00	48.34	54.43	52.84	53.62	55.47	59.43	57.38	55.86	60.71	58.18	78.67	80.33	79.46
S7 (S1–S6,S8–S15)	76.83	75.64	75.09	60.86	56.65	58.68	88.42	87.29	87.85	63.29	65.05	64.16	86.55	88.26	87.37
S8 (S1–S7,S9–S15)	47.24	47.00	49.25	66.00	63.64	64.80	60.00	61.29	60.64	84.86	85.84	85.34	93.53	91.73	92.61
S9 (S1–S8,S10–S15)	77.25	73.79	72.01	67.14	65.83	66.48	88.43	83.00	85.63	92.14	92.94	92.54	97.58	92.21	94.05
S10 (S1–S9,S11–S15)	63.23	61.00	57.41	58.14	52.31	55.07	66.18	64.29	65.22	58.86	61.04	59.93	85.52	83.24	84.30
S11 (S1–S10,S12–S15)	63.42	61.43	58.33	66.57	63.84	65.17	84.60	80.86	82.69	79.57	85.30	82.34	94.20	90.76	92.40
S12 (S1–S11,S13–S15)	64.45	63.64	62.60	66.57	58.84	62.47	68.22	71.14	69.65	62.57	68.98	65.62	85.39	85.22	85.20
S13 (S1–S12,S14–S15)	76.78	74.21	72.92	63.86	64.41	64.13	80.45	76.43	78.39	68.14	76.44	72.05	89.99	87.81	88.80
S14 (S1–S13,S15)	67.54	66.21	64.88	51.57	50.92	51.24	75.87	77.71	76.78	67.71	63.88	65.74	90.20	89.62	89.90
S15 (S1–S14)	56.43	56.07	54.81	48.71	47.63	48.16	50.37	48.43	49.38	44.00	45.29	44.64	88.15	87.94	88.00
**Average**	**62.74**	**61.65**	**60.14**	**60.60**	**57.64**	**59.05**	**70.74**	**71.44**	**71.05**	**68.35**	**72.12**	**70.14**	**89.08**	**88.02**	**88.46**

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
