# Peer review of "InstanceEasyTL: An Improved Transfer-Learning Method for EEG-Based Cross-Subject Fatigue Detection"

_sensors, 2020, doi:10.3390/s20247251_

Round 1
Reviewer 1 Report
This paper presents a novel classification algorithm of EEG data based on transfer learning models that can be used to automatically detect cross-subject fatigue while driving. Compared to other automatic classification algorithms the methodology proposed by the authors has a higher classification accuracy.
In my opinion, the use of transfer learning models to EEG data is an interesting novelty, and the proposed methodology seem to improve the original EasyTL algorithm. The paper however has some important drawbacks that could be attended on a review. I also acknowledge that I have limited experience with automatic classification algorithms, so I recommend that the proposed algorithm (sections 3 and 4) is revised in a deeper level by at least another reviewer with more expertise in this field.
My major concern about this paper is that the authors claim that their method is sensitive to the fatigue of the driver, but this fact is not proven anywhere in the study. The methodology developed by the authors seems to successfully detect changes in the brain activity patterns between two experimental conditions, but whether or not the differences on the brain activity are associated with fatigue is not demonstrated. It may be that the two experimental conditions require different levels of attention and/or working memory (i.e. cognitive load or cognitive effort), but it is not clear that these changes in the brain activity patterns are associated with fatigue (i.e. a performance decrease over time). Should the authors be unable to demonstrate that the neural differences between conditions are due to fatigue (and not cognitive effort), my recommendation is to rewrite all the sections of the paper in terms of “brain activity changes between two experimental conditions” and comment in the discussion whether these changes could be due to cognitive effort or fatigue.
It is also not clear why the authors picked the TAV3 and DROWN stages for comparison. This selection seems to be arbitrary and not based on an original hypothesis.
Other specific comments:
- Revise figure numbers and how they are referred in the manuscript. All figures have a wrong label (e.g. there is no figure 1 or figure 2 – the first figure is figure 3; and there are two figures 9).
- [line 3] Define cross-subject fatigue detection (if not in the abstract, please define this term in the introduction).
- [lines 70-75] Section 5 should also be presented.
- [line 79] I suggest to present the participants following these format: 15 subjects (X female, range [min-max] years, mean ± std = 26.8 ± 3.2 years).
- [lin 82] revise grammar in “all of them must be not drink alcohol”
- Table 1 can be suppressed and embedded on the text.
- [page 93-95] The “alert” and “vigilant” tasks require a deeper description. It is very difficult to understand these tasks with the current description. For example, it is not clear what are the stimulus intervals for each TAV, and what is actually the task.
- [page 97-98] I also find difficult to relate an ‘X’ on the screen with a complex environment like driving with traffic lights, pedestrians crossing the road, etc. – as the authors claim. This statement needs to be modified or supported with evidence.
- [page 109] How long does usually take participants to complete the rest of the stages?
- Are allowed participants to have breaks in between stages?
- If only TAV3 and DROWN stages are considered in the study, what is the purpose of the rest of the stages?
- [page 110] “Usually, the subjects will feel tired in the DROWN stage” – it is a missed opportunity to not have measured fatigue in all stages using other indicators (even via questionnaires). This is key aspect necessary to support the claim of their method being sensitive to fatigue.
- [page 114] When is the NASA-TLX questionnaire administered? At the very end of all stages? Is this questionnaire asking about their workload perception at the end or during each stage?
- [section 2.3] Was the ground in FCz? If so, please correct this in line 119. When were impedances meausured? The long duration of the test (2 hours) may have affected the impedances throughout the session. Is the band-pass filter that the authors refer a digital filter or is it the analogue filter of the EEG recording system. What EEG recording system was used? What electrode configuration was used? What electrode cap was used? What programming language was used to pre-process and process the EEG data?
- I encourage the authors to share their code to facilitate the implementation of their algorithm to interested users. Perhaps as supporting information.
- [Section 2.4] The statetment “Before feeding driving EEG data into…” is not clear. Please rewrite. Is this method applied to the EEG data from each stage? If so, please say this explicitely.
- [Figure 5a legend] Correct an error at the end - 1400 preprocessed EEG samples matrices each one is 61*100.
- [Section 3] Please define the source and target domains.
- [Section 4] Please change the title of this section from “Results and discussion” to “Results”, as section 5 presents the discussion.
- [line 207-208] “2 of 8 mental states… clearly distinguishable”. Evidence of this statement should be included in the paper without having to visit the references. How are they clearly distinguishable? Under what parameters?
Reviewer 2 Report
As the heading implies "An Improved Transfer Learning Method for EEG-based Cross-subject Fatigue Detection" by Hong Zeng et al, is concerned with improved detection of the fatigue during driving. This detection is vital, if road safety is to be improved. Which brings up the first and main question:
- How do the authors propose to achieve practical acquisition of the EEG signals during everyday driving? While it is not strictly withing the scope of the paper, the paper becomes largely irrelevant mathematical exercise if that kind of real time data acquisition is not possible or not feasible.
Additional remarks:
- Paper could benefit from thorough lingual corrections. At places (even in the abstract) the usage of the wording is borderline.
- Table 1 is misleading, since it leaves the impression that the actual car was used on actual track.
- Strange naming of WUP, PERFO and DROWN would benefit from better presentation of the meaning (even if the actual actions are presented).
- Fig. 5 a): probably it should be 61x100, and not the current 61x1400? Also letters on the figure and numbers in the text could be unified somehow.
- Fig. 8 and around (lines 176, 194): is "sample set" equal to the "domain"? - "source domain ...(also called ...)"
- Equation 11 could be presented and explained before table 3.
- Fig. 9 is presented twice.
Besides accuracy it would be interesting to see also the comparison of the computing power needed for presented and compared algorithms. Could it be easily embedded?
Round 2
Reviewer 2 Report
Authors have responded to the previous remarks. Unfortunately, not all the changes are for the benefit of the paper. It is still suggested that native English speaker should check the paper for lingual errors and omissions.
Few additional remarks:
“Helsinki Declaration of 1975, which was revised in 2008” is mentioned (line 81) – last revision is from 2013 and is the only official one.
Definition of the “TAV” should be clearer. Is it the “two extra kinds of stimuli tasks (TAV)“ in line 94 or something else? Even if it is, then it should certainly appear when the TAV is first mentioned (line91). Figure captions could be more descriptive in general.
